# Lumbar Pain in Patients with Multiple Sclerosis and Knowledge about Physiotherapeutic Methods for Combating Pain

**DOI:** 10.3390/healthcare11233062

**Published:** 2023-11-29

**Authors:** Martyna Odzimek, Waldemar Brola, Józef Opara

**Affiliations:** 1Doctoral School, Collegium Medicum, The Jan Kochanowski University, Żeromskiego 5, 25-369 Kielce, Poland; odzimek.martyna@onet.pl; 2Institute of Health Sciences, Collegium Medicum, The Jan Kochanowski University, al. IX Wieków Kielc 19A, 25-516 Kielce, Poland; 3Department of Physiotherapy, The Jerzy Kukuczka Academy of Physical Education, 40-065 Katowice, Poland; jozefopara@wp.pl

**Keywords:** low back pain (LBP), multiple sclerosis (MS), comprehensive rehabilitation, physiotherapy

## Abstract

Background: The purpose of this study was to evaluate the intensity and frequency of low back pain (LBP) in people with multiple sclerosis (PwMS) and patients’ knowledge of physiotherapeutic methods for combating LBP. Methods: This study included all MS patients attending consecutive follow-up visits for treatment related to MS between March and May 2023. Only current pain sensations in the lumbar spine were taken into account. The inclusion criteria were age 18–60 years, a definite diagnosis of MS according to the 2017 McDonald criteria, treatment with disease-modifying drugs (DMTs), and consent to participate in the study. This study was carried out using an original survey questionnaire and a Visual Analogue Scale. PwMS were divided into three age groups: 18–30 years, 31–50 years, and over 50 years. Results: Ninety PwMS (68 women and 22 men) were included in the study. The mean duration of the disease was 9.5 ± 4.9 years, and the mean EDSS was 3.5 ± 1.6. Most patients had a relapsing-remitting form of the disease. Overall, 68.9% of PwMS felt low back pain (*n* = 62). The relationship tested was statistically significant (*p* < 0.001), and the strength of the relationship was high (r_c_ = 0.695). The average level of low back pain among PwMS was 4.7 out of 10 on the VAS. The prevalence of LBP was higher in female patients (*p* < 0.001), patients with a secondary progressive form of MS (*p* < 0.001), and patients with a longer duration of disease (*p* < 0.05). The most widely used methods for treating LBP were kinesitherapy and manual therapy. Conclusions: LBP is common in patients with multiple sclerosis. Female sex, a secondary progressive form of MS, and a longer duration of disease increase the risk of LBP. It is important to implement properly planned physiotherapy activities and educate patients on how to combat LBP.

## 1. Introduction

Spinal pain (dorsalgia) is a term that describes a feeling of pain along the spine and paraspinal muscles that is either chronic (long-term occurrence for 3 months) or acute (short-term, intense, and lasting up to 3 months). It is one of the most common disorders of the musculoskeletal system in adults, and more than three quarters of the global population suffers from this disorder or will suffer from this disorder at least once in their lives. According to available epidemiological data, approximately 90% of cases involve pain of unknown etiology, and the potential hypothesis is largely related to overload, for example, heavy physical work, other comorbidities, a sedentary lifestyle, psychosocial disorders, or obesity. The problem can affect any of the five sections of the spine, but the vast majority of patients report problems in the lumbar-sacral and cervical areas. Furthermore, spinal disorders can be related to a lack of physical activity during the day (including the lack of spontaneous physical activity, for example, choosing the elevator instead of the stairs or using the car instead of walking), a sedentary lifestyle, and remote work, all of which significantly contribute to progressive pain [1,2,3].

Low back pain (LBP) is a multifactorial global disease and one of the three most common causes of disability in Poland (just after ischemic heart disease and stroke). According to data in 2017, a problem in the lower spine was reported in approximately 7.5% of the global population (approximately 577 million people) [4]. This health problem was estimated to have affected more than half of the surveyed respondents living in Poland in 2019, and pain was reported more frequently in men than in women (a difference of approximately 7%). It was reported much more frequently by people aged 15 to 49 years (more than three quarters of the group) than by people over 50. This health problem may be related to a greater workload in younger respondents than in the older group of respondents and also to socioeconomic or health factors. It should be noted that older people have more time to visit their doctors and to recover, which means that this health problem is recognized sooner and that the appropriate treatment can be implemented to improve the functional condition of the patient and reduce their pain [5,6].

A crucial issue worth mentioning in the context of low back pain is the number of hospitalizations due to this health problem. Hospitalizations are prevalent in women in retirement (over 60 years of age) and in men before retirement (below 65 years of age). Furthermore, in recent years, there has been an increase in hospitalizations due to lumbar spine pain among rural residents, both among women and men. The reasons for this phenomenon can be related to difficult working conditions (lack of appropriate equipment, lack of training, and lack of education in the field of physioprophylaxis) or socioeconomic conditions (lack of help from relatives, lack of money for treatment, and lack of time to look for the causes of the disease due to duties). The hospitalization rate for LBP in Poland is the lowest in Europe, but it should be noted that the average hospitalization time is statistically longer than in other European and non-European countries. The early implementation of physical therapy and preventive methods is important for the continued functioning of patients. A big problem in Poland is the lack of public health education and knowledge on how to counteract spine pain. It is estimated that half of the population with this health problem decides to seek the help of a specialist only when pain significantly prevents them from functioning in professional and/or social environments [7].

Multiple sclerosis is a progressive and inflammatory demyelinating disease of the nervous system that affects a large number of young people, regardless of location (both in developed and developing countries), making it a global disease to a large extent [8].

The cause of its occurrence is still unclear, but the analyzed literature indicates that great diversity and the influence of external factors contribute to the development of the disease. Currently, the number of people suffering from multiple sclerosis in the world is approximately 2.8 million, with an average incidence of 35.9 cases per 100,000 inhabitants. However, the highest incidence is recorded in European areas (incidence of 143 cases per 100,000 inhabitants). The most common information about multiple sclerosis is that this disease occurs more often in women (2:1) than in men. Patients with multiple sclerosis more often learn about their disease between the ages of 25 and 35 (average age of diagnosis is 32), and a diagnosis is associated with numerous changes in patients’ lives [9]. Symptoms that occur in people with multiple sclerosis can take different forms depending on the location of the lesions in the brain and/or spinal cord. Patients often struggle with problems in the upper and/or lower extremities in the form of paresis or spasticity. In addition, they experience changes in the perception of stimuli, such as vibration, pain, touch, or temperature. Problems with double vision, nystagmus, or damage to the trigeminal nerve (V) and/or facial nerve (VII) often occur. Furthermore, patients report sphincter disorders (urinary and/or fecal incontinence) and sexual problems. The course of this disease varies in each patient, so it is important to approach this health problem individually and to introduce appropriate therapy depending on the patient’s functional status, abilities, and general motivation [8,9,10,11].

Among the numerous symptoms and ailments, PwMS often complain of various types of pain. The frequency of different types of pain is estimated to range widely from 29 to 86% [12]. The most common type of pain is neuropathic pain [13], while LBP tends to occur in the later stages of the disease in older patients. Asymmetric posture and difficulty in ambulation can be predisposing factors for back pain. Lumbar spasticity is also a possible cause of lower back pain due to increased muscle tension and its effect on lumbar spine joints. Depending on the assessment methods used, the prevalence of LBP is estimated at 10–20% [14]. Most studies to date have generally focused on pain as a factor that significantly influences the quality of life of PwMS. Significantly less research has focused on LBP in MS and non-pharmacological ways of dealing with it. This type of research has not been conducted in Poland at all. Therefore, the authors decided to fill this research gap and assess the occurrence of LBP in Polish patients.

The purpose of this study was to evaluate the intensity and frequency of low back pain in patients with multiple sclerosis and to evaluate the knowledge of patients about physiotherapeutic methods for combating LBP.

## 2. Materials and Methods

This study included all MS patients attending consecutive follow-up visits related to DMT (disease-modifying therapy) between March and May 2023. Each patient had a definite diagnosis of MS according to the 2017 McDonald criteria [15]. All patients underwent neurological and general medical examination and were assessed using the EDSS (Extended Disability Status Scale) [16]. Then, an experienced physiotherapist specializing in the rehabilitation of PwMS conducted a focused interview regarding pain sensations, especially LBP, and conducted a VAS examination. Only current pain sensations in the lumbar spine were taken into account. Patients were divided into 3 age groups: 18–30 years, 31–50 years, and over 50 years. Due to the intensity of pain experienced, patients were further divided into the following groups:(1)0—without pain;(2)1–3—mild pain;(3)4–7—moderate pain;(4)8–9—severe pain;(5)10—unbearable pain [17].

Using a self-administered questionnaire prepared for this study, information regarding the basic personal data and the current situation of the patients (gender, age, place of residence, and education), information about their disease (subtype of MS, duration of disease, and EDSS score), and their knowledge about the methods of physiotherapy were collected.

The inclusion criteria included age 18–60 years, a definite diagnosis of multiple sclerosis according to the 2017 McDonald criteria, treatment with disease-modifying drugs (DMTs), and consent to participate in the study. The exclusion criteria of this study were age under 18 and over 60, a traffic accident within the last year, active cancer, and other serious additional diseases causing pain or requiring surgery of the lumbar spine. In addition, people were excluded from the study if they did not consent to participate, withdrew from the study, reported a deterioration of their health, or reported that unexpected accidents occurred.

Participants gave written informed consent to participate in the study, which was free and voluntary. They were informed why their consent was needed, and each stage of the study was explained. Each participant was informed about how this scientific study would be conducted and about the precautions that would be taken throughout. All participants were informed that they could stop the study at any time. Each participant obligatorily agreed to the use and analysis of anonymous data collected for this study. The researchers associated with this study had access to the research data. All research documentation containing personal data was protected. Each participant whose data were entered into the system was given a unique number, making it impossible to identify them. Participants also gave written consent to the publication of the research results in the form of a scientific article. This study was carried out according to the Declaration of Helsinki and in accordance with the applicable law in the Republic of Poland. The Ethics Committee did not require patient consent because this study was a pilot study, not a medical experiment. For this reason, we did not seek the approval of the Commission.

Statistical description techniques and the Shapiro–Wilk test of normality were applied for descriptions of the groups and the variables. The main research hypothesis was tested by comparing the average results of the two groups. The parametric Student’s *t*-test was applied since the groups were of similar sizes. To verify the secondary hypotheses, the same method was applied for the comparison of quantitative variables (e.g., age) between the groups, and a contingency table analysis with chi-square analysis was performed to compare the groups in terms of nominal and ordinal variables (e.g., gender or education). The magnitude and clinical relevance of the scores were evaluated based on effect size statistics. According to Cohen’s benchmarks, a value of 0 to 0.20 denotes a negligible effect size, a value of 0.21 to 0.5 denotes a small effect size, a value of 0.51 to 0.80 denotes a medium effect size, and a value > 0.80 denotes a large effect size.

All statistical analyses were performed using the STATISTICA 13.3PL software (StatSoft Poland, Kraków, Poland, 2017; www.statsoft.pl), and *p*-values < 0.05 were considered statistically significant.

## 3. Results

This study involved 90 people with multiple sclerosis aged 18 to 60 years (mean age 42.4 ± 12.6). The mean duration of disease was 9.5 ± 4.9 years, and the mean EDSS was 3.5 ± 1.6. Overall, 74.4% of the patients had a relapsing-remitting form, and 17.8% had a secondary progressive form. The respondents were divided into three age groups: 18–30 years old (57 people, 63.3%), 31–50 years old (23 people, 25.6%), and over 50 years old (10 people, 11.1%). The vast majority of the group was women (68 people, 75.6%), with a smaller group of men (22 people, 24.4%). Most of the respondents lived in small or large cities (*n* = 70, 77.8%), with a smaller group of people living in rural areas (*n* = 20, 22.2%). For both women (*n* = 53, 77.9%) and men (*n* = 17, 77.3%,), the most common place of residence was urban areas. Rural areas were inhabited by 22.1% (*n* = 15) of the women and 22.7% (*n* = 5) of the men. When asked about education, most respondents declared that they had completed secondary school (*n* = 36, 40%) and higher education (*n* = 29, 32.2%). The demographic and clinical data are presented in Table 1.

Most of the patients who participated in this study reported pain in the lumbar spine. LBP was felt by 68.9% of the patients (*n* = 62), while no problem was observed in 31.1% of the patients (*n* = 28). Significant differences were found depending on the gender of the respondents. Pain in the lumbar spine was reported more frequently by women (*n* = 56, 82.4% of the group) than by men (*n* = 6, 27.3% of the group). A total of 17.7% (*n* = 12) of women and 72.7% (*n* = 16) of men did not report this health problem. The relationship tested was statistically significant (*p* < 0.001, x^2^ = 23.558 > x^2^ = 10.828, df = 1), and the strength of the relationship was high (r_c_ = 0.695). The prevalence of low back pain was higher in patients with a secondary progressive form (*p* < 0.001) and a longer duration of disease (*p* < 0.05).

The results of this study showed significant associations between age (*p* < 0.001), secondary progressive form of MS (*p* < 0.001), female sex (*p* < 0.001), and LBP, with effect sizes in the range of 0.66–1.12 for Cohen’s values. Significant relationships between disease duration (*p* = 0.023) and EDSS score (*p* = 0.015) and LBP were also observed, while no significant associations were found between male sex, relapsing-remitting MS, and primary progressive MS. Detailed results are presented in Table 2.

In the entire study group (*n* = 90, 100%), more than 31.11% (*n* = 28) of the patients reported no recent pain in the lumbar spine. Mild and very severe pain were reported by the same number of patients (*n* = 16, 17.78%). Moderate pain was recorded in 26.67% of the patients (*n* = 24), and the most severe pain, preventing daily functioning and professional work, was recorded in 6.67% of the patients (*n* = 6). Significant differences in the intensity of lumbar spine pain were found across the sex groups. Women (*n* = 56, 82.35% of the female group) reported lumbar spine pain (from 1 to 10 on the VAS) much more frequently than men (*n* = 6, 27.27%). Pain was not observed in 17.65% of the women (*n* = 12) and 72.73% of the men (*n* = 16). Women most frequently reported pain of moderate intensity (*n* = 23, 33.82%) and mild intensity (*n* = 15, 22.06%). Very severe pain and the most intense pain affected 19.12% (*n* = 13) and 7.35% (*n* = 5.56%) of the women, respectively. Men most often reported severe pain (*n* = 3, 13.63%), and mild, moderate, and very severe pain was observed in only one man (*n* = 1, 4.55%). The average level of low back pain among these MS patients was 4.7 out of 10 on the VAS. The relationship tested was statistically significant (*p* < 0.001, x^2^ = 24,919 > x^2^ = 18,467, df = 4), and the strength of the relationship was medium (r_c_ = 0.466). Detailed results are presented in Table 3.

The last part of this study involved evaluating the knowledge of patients with multiple sclerosis about physiotherapeutic methods used to treat or reduce lumbar spine pain (multiple-choice question). A total of 65.55% of patients with multiple sclerosis believed that the best and most effective physiotherapeutic treatment for lumbar spine pain is manual therapy. Kinesiotherapy also ranked very high, with a score of 66.33%. Much less frequently, the respondents mentioned deep tissue massage (30.00%), trigger-point therapy (26.67%), and kinesiotaping (24.44%). The respondents considered ultrasound treatment (2.22%) and heat therapy (8.89%) to be the least effective methods to combat lumbar spine pain. Electrotherapy (16.68%) and laser therapy (12.22%) also had low values. Concerning the use of physiotherapeutic treatments for lumbar spine pain, the respondents had most often used kinesitherapy (71.11%), manual therapy (46.67%), electrotherapy (43.33%), or laser therapy (36.67) at least once. The methods that were chosen least frequently were trigger-point therapy (10.00%), kinesiotaping (14.44%), and deep tissue massage (16.67%). The least frequently chosen physical treatments were ultrasound treatment (12.22%) and heat therapy (13.33%). The results are presented in Table 4.

## 4. Discussion

Lower back disorders are a civilized and global problem that affects the vast majority of the population at least once in their life, regardless of sex or age. Differences in the appearance of this health problem are evident, especially among professionally active people, and may be related to a sedentary lifestyle or a lack of physical activity in daily life [18]. The main objective of the systematic review conducted by Meucci’s team was to assess the prevalence of chronic low back pain worldwide, depending on the age and gender of the patients. An extensive analysis of the 28 studies included in the review found a significant relationship between age and lumbar spine pain (19.6% of people aged 20–59 years and 4.2% of people aged 24–39 years). This analysis also supports the theory that lumbar spine pain is more common in women and gradually increases until the age of 60 [19]. Important issues in the treatment of people with low back pain are detailed diagnostics and conservative treatment. Scientific research points to the importance of using ‘yellow, blue and/or black flags’ to assess disability risk. To a large extent, treatments use non-pharmacological elements in the form of physiotherapy, targeted and appropriately selected physical activity, or additional patient education. It is worth mentioning that the use of pharmacological methods is still widely debated [20].

When it comes to the presence of pain in patients with multiple sclerosis, scientific studies and clinical reports often show an inconsistent picture. Pain is often neuropathic and may affect more than 80% of patients, especially in the form of pain in the upper and/or lower limbs, neuralgia, spinal pain syndromes (especially cervical or lumbar), or headaches [13]. A review of scientific studies shows that patients have problems with back pain, and the prevalence ranges from a few percent to as much as 50%. In this literature analysis, a major problem is the lack of an unambiguous indication of the location of the pain segment, which may significantly affect the results of the analysis [14].

In previous studies, the average age of patients with multiple sclerosis participating in studies related to lumbar spine pain ranged from 30.4 years to as much as 54.9 years [21,22,23,24,25]. The authors’ research results are in a similar age range (42.4 ± 12.6). Most of the studies included numerous groups of patients, most of whom were women, which is very similar to the authors’ study (75.56% of the participants were women) [21,22,23,24,25,26,27]. The most common participants in research were people with relapsing-remitting multiple sclerosis, which accounted for 23.7% to 93% of the participants across studies [21,23,26,27,28]. Only in one study did the largest group consist of people with secondary progressive multiple sclerosis [25]. In our own research study, the largest group included patients with relapsing-remitting (73.3%) and secondary progressive multiple sclerosis (18.9%). The smallest group included patients with primary progressive multiple sclerosis (7.8%). In previous studies, the mean EDSS score ranged from 2.0 to 6.0 [23,25,26,27,28], and the mean duration of disease ranged from 2.2 to 19.3 years [21,22,23,24,25,26,27,28]. In this study, the mean duration of disease was 9.5 ± 4.9 years, and the mean EDSS was 3.5 ± 1.6.

The examination of lumbar spine pain was carried out using various methods and tools. Authors and researchers most often used numerical scales, questionnaires, and proprietary questionnaires to assess lumbar spine pain in people with multiple sclerosis. A Visual Analogue Scale was used in this study, as used by the authors of the abovementioned studies [23,26,28]. In addition, this study was extended to include the EDSS scale [22,23,25,26,28]. Previous studies had used the Nordic Musculoskeletal Questionnaire [24,27], painDETECT [27], and a Numerical Rating Scale [22,24,25]. Despite the discrepancies in the research tools used, problems in the lower spine were reported in 26.6% to 52.4% of patients with multiple sclerosis [21,23,25,26,27,29]. A significantly higher result (68.89%) was obtained by the authors of a previous study. It is worth mentioning that researchers in Lithuania came to different conclusions in which lumbar spine pain was more common in the control group (40.0%) than in the study group (21.7%) [22]. In addition, one of the studies included other segments of the spine, and respondents were much less likely to report pain in the thoracic (2.1%) or cervical (3.2%) spine [25]. The intensity of pain was reported at various levels, from mild to moderate to severe [23,26,29]. The average value ranged from 5.6 to 6.75 [23.26]. Similar results were noted by the authors of the abovementioned study, where pain in the lumbar spine was reported by 68.89% of the respondents, and the average pain intensity was estimated to be 4.7. In addition, the results showed a statistically significant relationship between multiple sclerosis and pain intensity. People with musculoskeletal pain in the lower back had a lower EDSS score than those diagnosed with neuropathic pain [27]. According to Drulovic’s research, patients who experienced pain were older and presented with a higher disability score according to the EDSS scale [29]. In our own research, we used the EDSS scale, showing an average of 3.2 ± 1.2 in women and 3.7 ± 1.8 in men. Importantly, the incidence of low back pain was higher in patients with secondary progressive disease (*p* < 0.001) and a longer duration of disease (*p* < 0.05). The differences between the groups may be due to different conditions, as identified in the Bento et al. study. Low back pain has been associated with older age, low levels of education, hypertension, and smoking in men. However, occupational and ergonomic factors are a more common cause of pain in women [6]. In addition, it is worth mentioning that pain has a significant impact on the quality of life of patients with multiple sclerosis. Quality of life is significantly reduced and is associated with depression, fatigue, and lack of physical activity in people with multiple sclerosis. [30]. However, balance problems, sensory disorders, and difficulty walking have the greatest impact on quality of life. Lack of independence may cause additional behavioral disorders [31]. Sociodemographic factors also have a significant impact on the functioning of patients with multiple sclerosis [32], as well as family situation and occupational factors [30]. Sleep disorders may also have a significant impact, but available reports indicate that they may be associated with other diseases as well, not only multiple sclerosis [33].

Most guidelines for the treatment of low back pain recommend maintaining physical activity and selecting the parameters, duration, and type of exercise depending on patients’ functional status [34,35]. The American College of Occupational and Environmental Medicine (ACOEM) recommends the use of aerobic exercise to increase the body’s cardiovascular fitness [36]. Studies confirm the positive impact of such exercises not only on reducing pain but also on the mental health of patients [18,37]. In addition, in the treatment of people with low back pain, it is important to use stretching exercises (3–5 times a day) [18,36] to restore mobility of the lumbar spine and other segments. It is important to use strengthening exercises to improve and increase the endurance of the core muscles [18,36]. In addition, lumbar spine stabilization exercises and general rehabilitation exercises may be of significant importance in the treatment of LBP [18,38]. It is worth emphasizing, however, that there are no significant differences regarding which muscle is stabilized [39]. The literature also notes the positive effects of yoga [36,37,40,41], Tai Chi [36,40,41], and Pilates [41,42]. The use of Pilates as a rehabilitation tool is effective in achieving the desired results, especially in the area of pain and disability reduction [43]. Stretching exercises (especially in the case of significant limitations in the mobility of the lumbar spine) or exercises that strengthen the abdomen (as the only form of activity) [36] are not recommended. Scientists indicate that the use of relieving exercises in water may be of significant importance in the recovery of people with low back pain and comorbidities [36].

In the treatment of patients with LBP, it is recommended to use techniques from the field of manual therapy, in particular manipulation [40], mobilization [18,44], or soft tissue techniques [18,34,41,45,46]. The use of traction in the treatment of low back pain is not recommended [34,37,41,45,46]. In addition, opinions regarding the use of therapeutic massage in the rehabilitation process of people with low back pain are divided. This form of physiotherapy is not recommended by Kreiner [37], while other studies indicate a positive effect on the body, especially in terms of muscle relaxation [18,36,47]. The use of trigger-point therapy may be more effective than the use of pressure in random areas combined with superficial massage [48].

Despite the use of numerous physiotherapeutic treatments, it is worth mentioning treatments that are not recommended in the case of low back pain. It is not recommended to use stabilizing belts and corsets [41,45,46], foot orthoses [34,45,46], ultrasound [34,35,37,41,45], transcutaneous electrical nerve stimulation (especially as the only procedure performed) [35,37,41,45,46], interference therapy [45,46], kinesiotaping [35], shortwave diathermy [35,36], laser therapy [35], magnetotherapy [35], or cryotherapy [35,36].

The interdisciplinary rehabilitation process should include cognitive behavioral therapy [41] and be supplemented with an appropriate model of patient education [34,35,37,45,46] and instruction in continuing physical activity [34,35,46]. A return to work and a social life are important [36,41]. Disability caused by lumbar spine pain can have a significant impact on patients’ mental state and cause problems in returning to daily life or work. In addition, the results of previous studies show that anxiety can have a significant negative impact on the outcomes of chronic low back pain syndromes [49].

An in-depth analysis of the available scientific literature allowed us to find information on physiotherapeutic methods used for treating low back pain in people with multiple sclerosis. The Al-Smaidi study involved 15 people who were divided into three groups and underwent appropriate therapy with the use of TENS currents. The first group consisted of people who received low-frequency TENS currents (4 Hz, 200 μs), the second group consisted of people who received high-frequency TENS currents (110 Hz, 200 μs), and the third group (control) did not receive electrotherapeutic stimuli. These patients were treated for a maximum of 10 weeks, and the frequency of treatments was twice a week for 45 min. Test scores were measured using the Roland Morris Disability Questionnaire, Short Form-36, the McGill Questionnaire, and a Visual Analog Scale. The researchers concluded that the use of TENS currents was effective in the treatment groups when compared to the control group, particularly in reducing VAS scores. However, despite this novel approach, the results were statistically insignificant [50]. The above research model was reproduced after 3 years by the same team (with minor personnel changes). To obtain reliable results, the size of each group was increased to 30 people, and the parameters used during TENS therapy did not change. The outcomes were measured using a Visual Analogue Scale, the McGill Visual Questionnaire, the Roland Morris Disability Questionnaire, the Barthel Index, and the Rivermead Mobility Index. As in the previous work, the team did not obtain statistically significant results [51]. However, it is important to mention that physiotherapy, especially kinesitherapy, is important in the treatment of patients with multiple sclerosis and lumbar spine pain. In order to improve the patients’ condition, it is important to select proper exercises and spontaneous physical activity, most often in the form of aerobic, resistance, stretching, or balance exercises. Exercise is one of the safest forms of rehabilitation, and scientific evidence supports its effectiveness in improving fitness, efficiency, and quality of life [52]. It is worth noting that the introduction of spinal stabilization training may be important in improving stability and balance in people with multiple sclerosis. In a previous study, its effectiveness was noticeable after an 8-week rehabilitation program in more than 62% of patients [53]. The Pilates method may bring significant benefits in the treatment of people with multiple sclerosis and concomitant pain of the lumbar spine. This method focuses on restoring the proper stability, strength, and flexibility of muscles; improving muscle control; and normalizing breathing. A systematic review showed that the use of the Pilates method results in a significant improvement in gait, muscle strength, and balance [42]. In addition, it may reduce the feeling of fatigue, although the results of previous studies are inconclusive on this issue. Exercise in the form of Pilates can affect the quality of life of patients with multiple sclerosis, but additional research is still needed in this area [42,54]. There are scientific studies that support the theory that a progressive type of multiple sclerosis and impaired vision can increase the risk of back pain. The results of these studies may lead to a breakthrough in the treatment of lower back pain and highlight the importance of the prevention of vision disorders in people with multiple sclerosis [55,56].

The high incidence of pain reduction indicates the importance of regular physiotherapeutic treatments [26]. Important effects in the treatment of people with LBP include a reduction in pain, an improvement in cardiorespiratory efficiency, increased mobility of the trunk, and an improvement in movement control. It will also be important to educate patients about pain management options and to provide home recommendations [18].

In our study, the main limitations were the small size of the study group and the lack of a control group. This study also had a limited time frame, and not all patients agreed to participate in it. These were only patients taking DMTs, while some untreated patients were not included in the study. The inclusion criterion of age up to 60 years also excluded several patients from the study. The conditions of some people did not allow them to participate in the study. Moreover, people with lumbar spine injuries had to be excluded from the study, which made it impossible to collect a sufficiently large sample size (especially patients with a secondary progressive form of MS). Moreover, the assessment of pain was limited only to a VAS; other more extensive scales, such as the Self-Complete of Leeds Assessment of Neuropathic Symptoms and Signs (S-LANSS), the McGill–Melzack Questionnaire, the Brief Pain Inventory (Short Form), the Oswestry Disability Index (ODI), the painDETECT, the Quebec Back Pain Disability Scale, the Roland–Morris Disability Questionnaire, and the Laitinen Scale, could provide a broader view of the nature of LBP. This study focused on estimating the incidence of LBP in MS patients, ignoring the aspects of the impact of pain on quality of life. This topic, according to the authors, exceeded the scope of this publication, and it was decided that it would be the subject of another report.

## 5. Conclusions

According to population-based studies, the annual incidence of a first episode of LBP ranges from 6.3% to 15.4%, while the cumulative incidence of a first or any episode ranges from 1.5% to 36% [57]. Multiple sclerosis is a chronic and incurable disease accompanied by various complications. Due to the weakening of muscle strength and increased muscle tension, as well as disturbances in gait statics, patients often complain of LBP. This applies especially to patients who have been ill for a longer period of time and with secondary progressive forms of MS. Most (68.9%) of the PwMS in our study reported low back pain. So far, there are no precise epidemiological data on the incidence of LBP in PwMS and no clear management guidelines. Pharmacological treatment rarely brings relief from the symptoms of LBP. Patients most often use various forms of physiotherapy. In our study, 65.6% of PwMS believe that the best and most effective method of physiotherapy for LBP is manual therapy. In the case of LBP, the participants also used kinesitherapy (71.11%), electrotherapy (43.33%), or laser therapy (36.67%) at least once. There is a need to prepare guidelines on which rehabilitation methods are most effective in the treatment of LBP in MS. An analysis of the available literature and our own research allow us to conclude that there is a relationship between LBP and multiple sclerosis. It is important to implement properly planned physiotherapy activities and educate patients on how to combat lumbar spine pain.

## Figures and Tables

**Table 1 healthcare-11-03062-t001:** Demographic and clinical characteristics of patients.

Characteristics/Group	Male, *n* = 22	Female, *n* = 68	Total, *n* = 90
Age (years), *n* (%)			
18–30	15 (16.7)	42 (46.6)	57 (63.3)
31–50	5 (5.6)	18 (20.0)	23 (25.6)
50+	2 (2.2)	8 (8.9)	10 (11.1)
Place of residence, *n* (%)			
Small/big city	17 (18.9)	53 (58.9)	70 (77.8)
Village	5 (5.6)	15 (16.7)	20 (22.2)
Education, *n* (%)			
Primary	10 (11.1)	15 (16.7)	25 (27.8)
Secondary	8 (8.9)	28 (31.1)	36 (40.0)
University	4 (4.4)	25 (27.8)	29 (32.2)
Subtypes of disease course, *n* (%)			
Relapsing-remitting	15 (16.7)	52 (57.8)	67 (74.4)
Primary progressive	2 (2.2)	5 (5.6)	7 (7.8)
Secondary progressive	5 (5.6)	11 (12.2)	16 (17.8)
EDSS score (mean ± SD)	3.2 ± 1.2	3.7 ± 1.8	3.5 ± 1.6
Duration of disease (years, mean ± SD)	9.7 (5.1)	9.3 (4.8)	9.5 (4.9)

Note: EDSS—Expanded Disability Status Scale; SD—standard deviation.

**Table 2 healthcare-11-03062-t002:** Lumbar pain in patients with multiple sclerosis.

Patients	Patients with LBP *n* = 62	Patients without LBP *n* = 28	ES Cohen’s d	*p*-Value
Gender, *n* (%)				
Male	6 (9.7)	16 (57.1)	NS	0.45
Female	56 (90.3)	12 (42.9)	0.95	**<0.001**
Age (years, mean ± SD)	46.5 (13.4)	36.7 (10.4)	0.84	**<0.001**
Duration of disease (years, mean ± SD)	12.1 (4.4)	7.1 (3.2)	0.72	**0.023**
Clinical course, *n* (%)				
Relapsing-remitting	42 (67.4)	24 (85.7)	NS	0.085
Primary progressive	4 (6.4)	3 (10.7)	NS	0.595
Secondary progressive	16 (25.8)	1 (3.6)	1.12	**<0.001**
EDSS score (mean ± SD)	4.6 (2.2)	2.4 (1.2)	0.66	**0.015**

Note: EDSS—Expanded Disability Status Scale; ES—effect size; LBP—low back pain; NS—not significant; SD—standard deviation; statistically significant differences are shown in bold.

**Table 3 healthcare-11-03062-t003:** Intensity of pain in lumbar spine (Visual Analogue Scale).

Intensity of Pain	Female		Male		Total, *n* (%)
	*n,* % of the Entire Group	% of the Female Group	*n,* % of the Entire Group	% of the Male Group	
0	12 (13.33)	17.65	16 (17.78)	72.72	28 (31.11)
1–3	15 (16.67)	22.06	1 (1.11)	4.55	16 (17.78)
4–7	23 (25.56)	33.82	1 (1.11)	4.55	24 (26.67)
8–9	13 (14.44)	19.12	3 (3.33)	13.63	16 (17.78)
10	5 (5.56)	7.35	1 (1.11)	4.55	6 (6.67)
Total, *n* (%)	68 (75.56)	100	22 (24.44)	100	90 (100)

**Table 4 healthcare-11-03062-t004:** The most effective physiotherapeutic methods used at least once in a lifetime for the treatment of lumbar spine pain.

Physiotherapeutic Method	The Most Widely Used Method *n*, %	I Have Used at Least Once in My Life *n*, %
Manual therapy	59 (65.55)	41 (46.67)
Kinesitherapy	57 (63.33)	64 (71.11)
Deep tissue massage	27 (30.00)	15 (16.67)
Trigger-point therapy	24 (26.67)	9 (10.00)
Kinesiotaping	22 (24.44)	13 (14.44)
Electrotherapy	15 (16.68)	39 (43.33)
Laser therapy	11 (12.22)	33 (36.67)
Thermal therapy	8 (8.89)	12 (13.33)
Ultrasounds	2 (2.22)	11 (12.22)

## Data Availability

The data presented in this study are available from the first author upon request.

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
