# Peer review of "Lumbar Pain in Patients with Multiple Sclerosis and Knowledge about Physiotherapeutic Methods for Combating Pain"

_healthcare, 2023, doi:10.3390/healthcare11233062_

Round 1

Reviewer 1 Report

Comments and Suggestions for Authors

Review of the manuscript „Lumbar Pain in Patients with Multiple Sclerosis and Knowledge about Physiotherapeutic Methods of Combating”

The topics covered in the manuscript seem important, but the study has a number of shortcomings. Main remarks:

1.      Introduction

·        The introduction consists only of the characteristics of spine pain and multiple sclerosis. It is also worth justifying why the authors want to deal with the topic, whether there is already any research on this topic, and what gaps there are in the research.

·        Lines 78 – 84 – data inconsistent with the cited source [9]

·        Line 80 – 2.900,00 - the number is written incorrectly

2.      Poorly written chapter "Material and methods"

·        No description of how participants were recruited

·        No description of the methods used (only those mentioned), no reference to the literature

·        Line 101 - Numerical pain intensity scale is not the same as the Visual Analogue Scale

·        Did the question about pain concern current pain or pain that had also occurred in the past (e.g. during the last year)?

·        Detailed characteristics of the group should be included in the results section

·        Statistics description - incomplete (The chi-square test was probably not used to analyze quantitative data (e.g. age, duration of the disease, EDSS)?) + the description should be placed at the end of the "Material and methods" section, not in its middle.

3.      Results

·        Line 137 – p=0.0000012 should be written as p<0.001

 l. 167 – p=0.00005 should be written as p<0.001

·        Tab. 2 - analyzes are performed for extremely small groups (secondary progressive without LBP - 1 person, 3 people)

·        There is no legend explaining the abbreviations below table 2. Abbreviations should also be expanded when they first appear in the text

·        Liness 144-150 – a description of the pain scale (numerical? Or VAS?) should be placed in the methods section

·        Lines 152-153 - study (n=90, 100%), more than 31.11% (n=28) - repetition

4.      Discussion

·        it is worth describing the limitations of the study (there are many of them)

Reviewer 2 Report

Comments and Suggestions for Authors

Reviewer's Report:

Title: Lumbar Pain in Patients with Multiple Sclerosis and Knowledge About Physiotherapeutic Methods of Combating

Abstract:

The abstract provides a concise summary of the study, but it lacks specific details regarding the methodology and results. It could benefit from including more specific information about the sample size, key findings, and implications of the study. Additionally, the abstract should mention the limitations of the research to provide a more balanced overview.

Introduction:

1. The introduction provides some background information on spinal pain and multiple sclerosis but lacks clarity and structure. It is important to organize the information logically and provide a clear rationale for the study.

2. The introduction should clearly state the research objectives and hypotheses to guide the reader's understanding of what the study aims to achieve.

3. There is a need for proper citation and referencing of sources used for the background information to support the statements made.

Methods:

1. The Methods section lacks detailed information about the survey questionnaire and the Visual Analogue Scale (VAS) used for pain intensity assessment. Provide more information on the questions asked in the questionnaire and how the VAS was administered.

2. The inclusion and exclusion criteria for selecting patients with multiple sclerosis should be explicitly stated.

3. Ethical considerations should be discussed more comprehensively, including how informed consent was obtained and ethical approval, if applicable.

4. While you mentioned that statistical analysis was performed, it would be helpful to include more details about the specific statistical tests used and why they were chosen.

RESULTS AND DISCUSSION

1. Lack of Control Group: The manuscript mentions the absence of a control group, which is a significant limitation. Without a control group, it is challenging to draw strong conclusions about the relationship between LBP and multiple sclerosis. The absence of a control group makes it difficult to compare the prevalence of LBP in PwMS with the general population or other relevant groups.

2. Inconsistent Literature Comparison: The manuscript attempts to compare its findings with previous studies, but the comparisons are not always consistent. For example, the study mentions that women reported more LBP than men, which contradicts some previous studies. The manuscript should provide a more comprehensive review of the literature and clarify any discrepancies.

3. Limited Discussion of Pain Intensity: The manuscript briefly discusses the intensity of LBP using the Visual Analog Scale (VAS), but it does not delve into the clinical implications of these pain intensity levels. It would be beneficial to provide more context on how the reported pain levels might impact the quality of life and daily functioning of PwMS.

4. Statistical Significance: While the manuscript reports statistical significance in some findings, it does not provide effect sizes or clinical significance. It would be helpful to include effect size measures to determine the practical importance of the observed differences.

5. Physiotherapy Methods: The manuscript discusses the preferred physiotherapy methods chosen by PwMS for LBP treatment. However, it does not offer insights into the effectiveness of these methods or whether they are recommended by healthcare professionals. Providing more information on the evidence-based effectiveness of these methods would enhance the discussion.

6. Conclusions and Implications: The conclusions should be more specific and discuss the implications of the findings for clinical practice and future research. How can these findings inform the management of LBP in PwMS, and what further research is needed to address existing gaps in knowledge?

7. Grammar and Clarity: The manuscript would benefit from further proofreading and clarity improvements. Some sentences are lengthy and complex, making it challenging for readers to follow the arguments.

Comments on the Quality of English Language

The manuscript would benefit from further proofreading and clarity improvements. Some sentences are lengthy and complex, making it challenging for readers to follow the arguments.

Round 2

Reviewer 1 Report

Comments and Suggestions for Authors

All comments were taken into account in the new version of the work and doubts were clarified.

Reviewer 2 Report

Comments and Suggestions for Authors

The authors have conscientiously and successfully dealt with the raised queries, leading to a significant improvement in the manuscript's overall quality. Consequently, I am delighted to recommend the acceptance of the manuscript in its present state.